# STAT5 as a Key Protein of Erythropoietin Signalization

**DOI:** 10.3390/ijms22137109

**Published:** 2021-07-01

**Authors:** Zuzana Tóthová, Jana Tomc, Nataša Debeljak, Peter Solár

**Affiliations:** 1Institute of Medical Biology, Faculty of Medicine, P.J. Šafárik University in Košice, 04011 Košice, Slovakia; zuzana.tothova1@student.upjs.sk; 2Medical Centre for Molecular Biology, Institute of Biochemistry and Molecular Genetics, Faculty of Medicine, University of Ljubljana, 1000 Ljubljana, Slovenia; jana.tomc@mf.uni-lj.si (J.T.); natasa.debeljak@mf.uni-lj.si (N.D.)

**Keywords:** STAT5, erythropoietin, erythropoietin receptor

## Abstract

Erythropoietin (EPO) acts on multiple tissues through its receptor EPOR, a member of a cytokine class I receptor superfamily with pleiotropic effects. The interaction of EPO and EPOR triggers the activation of several signaling pathways that induce erythropoiesis, including JAK2/STAT5, PI3K/AKT, and MAPK. The canonical EPOR/JAK2/STAT5 pathway is a known regulator of differentiation, proliferation, and cell survival of erythroid progenitors. In addition, its role in the protection of other cells, including cancer cells, is under intense investigation. The involvement of EPOR/JAK2/STAT5 in other processes such as mRNA splicing, cytoskeleton reorganization, and cell metabolism has been recently described. The transcriptomics, proteomics, and epigenetic studies reviewed in this article provide a detailed understanding of EPO signalization. Advances in this area of research may be useful for improving the efficacy of EPO therapy in hematologic disorders, as well as in cancer treatment.

## 1. STAT5 Signaling and Erythropoiesis 

In classical erythropoiesis, erythropoietin (EPO) binds to an EPO receptor (EPOR) homo-dimer, activating EPO/EPOR signal transduction. EPO has two non-identical binding sites on EPORs: a high-affinity G151 (nano-molar) on the first and a low-affinity R103 (micro-molar) on the second receptor. The main signal cascades activated by EPO are JAK2/STAT5 cascade, phosphatidylinositol 3-kinase (PI3K) cascade, RAS/MAP kinase cascade, and protein kinase C (PKC) cascade [1]. 

Janus kinase 2 (JAK2) is a non-receptor tyrosine kinase that represents a pivotal role in signal transduction via its association with EPOR. There are diverse models of EPO/EPOR signaling. One supposes that EPO binding stimulates dimerization of the EPOR, which is important for signalization [2]. Another suggests the existence of a pre-assembled EPOR homodimer and JAK2 attached via the box1-2 motif to each monomer of EPOR. EPO induces a conformational twist of the two EPOR monomers, resulting in cross-inhibition of JAK2 kinase domains and JAK2 phosphorylation [3]. Investigations by McGraw et al. [4] have shown that EPO also induces EPOR translocation to membrane rafts, accompanied by the incorporation of JAK2, STAT5, and tyrosine-protein kinase LYN. 

Signal transducer and activator of transcription 5 (STAT5) refer to two proteins that share 94% structural homology and are transcribed from separate genes, STAT5A and STAT5B. However, they are both essential in erythroid differentiation [5]. STAT5 protein is recruited to phosphorylated tyrosine residue 344 (pY343) of the EPOR via its SH2 domain [6] and is phosphorylated by JAK2, although non-tyrosine-containing sequences in the cytoplasmic region of EPOR can also stimulate STAT5 [7]. Subsequently, STAT5 undergoes a conformational change and forms parallel dimers via the SH2 domain. Such conformational reorganization is possible due to the flexible linkers that connect the core fragment of STAT5 with its N- and C-terminus [8]. Established STAT5 dimers are recognized by importins—importin α3 and β1, respectively—which allow them to translocate into the nucleus [9]. STAT5 binds as a dimer or tetramer to palindromic gamma-activated sequences called GAS0 motifs (TTCYXRGAA) in promoters and enhancers of target genes, resulting in their transcription [10]. The proto-oncogene tyrosine-protein kinase Src (SRC) can directly phosphorylate tyrosine residue 694 (pY694) of STAT5 and so contribute to EPO-induced signal transduction via STAT5 [11]. FYN, an Src-family tyrosine kinase, also participates in the EPO signaling pathway, since Fyn^−/−^ mice exhibit reduced tyrosine phosphorylation of EPOR and decreased STAT5 activity. The importance of FYN in erythropoiesis is also supported by the blunted responsiveness of Fyn^−/−^ mice to stress erythropoiesis [12]. Another member of the Src-family tyrosine kinases, LYN, is involved in tyrosine phosphorylation and activation of STAT5, mainly on tyrosine residue 694 (pY694) [13]. Besides, CRK-like adapter protein (CRKL) seemed to have a role in STAT5 activation since direct binding of CRKL to STAT5 was observed after EPO stimulation in an (EPO)-dependent cell line UT-7/EPO [14]. 

Recently, 121 new signals and/or EPO/EPOR/JAK2 target proteins, together with their EPO-modulated domains and phosphosites (e.g., pT349, pS358 and others), were identified. These included actin assemblage modifiers with adaptors DLG-1, DLG-3, WAS, WASL, and CD2AP, and erythroid cytoskeletal targets such as spectrin A, spectrin B, adducin 2, and glycophorin C. Metabolic regulators such as aldolase A, pyruvate dehydrogenase α1, and thioredoxin-interacting protein were also found as an EPO target set [15]. Furthermore, the same authors identified several novel EPO-modified kinases and phosphatases, including membrane palmitoylated protein 1, guanylate kinase 1, pseudopodium enriched atypical kinase 1, AP2 associated kinase 1, protein tyrosine phosphatase receptor type A, phosphohistidine phosphatase 1, tensin 2, ubiquitin-associated and SH3 domain-containing B (UBASH3B), and protein tyrosine phosphatase non-receptor type 18 (PTPN18). Due to its high expression in hematopoietic progenitors, PTPN18 has been further investigated, and it was revealed that PTPN18 promotes ERK1/2, AKT, STAT5, and JAK2 signals for EPO-dependent hematopoietic cell growth (Figure 1, Table 1) [15].

## 2. The Interplay of EPO Signaling, Transcription Patterns, and Epigenetics

EPO-dependent STAT5 target genes involved in erythropoiesis are well-known. These are the following: Bcl-XL, Bcl2l1, Pim1, Cish, Socs3, Podocalyxin, Maltr3, Chac1, Ccrn4l, Socs2, Tnfr-sf13c, Lrp8, Cmtm6, Gdf3, Oncostatin-m, Rpl12, Lyl1, Gas5, Pim3, Bim, Trb3, Serpina-3G, Irs2, Suv420h2, Gypc Cdc25a, Btg3, p27-kip1, Trb2, Klf3, Cyclin-G2, Cyclin-D2, CyclinB1-IP-1, Spi2A, and others [20,24,25]. Some of these STAT5 target genes are negative regulators of STAT5 signaling. For example, proto-oncogene serine/threonine-protein kinase Pim-1 (PIM1) suppresses cytokine signaling 1 (SOCS1) and SOCS3. PIM1, however, does not bind directly to STAT5. It interacts with SOCS1 and SOCS3 and potentiates their inhibitory effects on STAT5, most likely via phosphorylation-mediated stabilization of the SOCS proteins [24]. STAT5 activity is also regulated via the inhibitory mechanism of tyrosine-protein phosphatase non-receptor type 6 (SHP-1, encoded by PTPN^6 gene) which associates with the tyrosine-phosphorylated cytoplasmic domain of the EPOR. Moreover, SH2B adapter protein 3 (Lnk—signal transduction protein) involved in the signaling of cytokine receptors inhibits erythropoiesis and EPO-dependent JAK2 activation together via downstream signaling pathways [26]. The authors demonstrated that Lnk blocks STAT5 pathways induced by EPO in primary erythroblasts. In addition, SH2B adapter protein 2 suppresses EPO-induced STAT5 activation through the binding and a masking effect on STAT5 docking sites of EPOR. Negative feedback on EPO-induced signaling also provides CISH protein (the CISH gene is a direct target of STAT5), which binds activated pY401 of EPOR. Similar to other SOCS family members, CISH forms a complex with elongin B, elongin C, cullin5, and Rbx2. Such complex functions as an E3 ubiquitin ligase for ubiquitination and subsequent proteasomal degradation of EPOR [20]. SOCS3, however, provides a slightly different feedback dynamic compared to that of CISH. It also contains an N-terminal KIR peptide that directly affects STAT5 binding via JAK2. Gillinder et al. [20] suggested another STAT5-dependent negative feedback loop of EPOR downregulation. They reported that Clint1 (EpsinR) might enhance the internalization and/or endocytosis of EPOR through the association with clathrin-coated vesicles (Figure 1, Table 1). Other than Socs1 and Socs3, Socs2, Spred1, Spred2, and Eaf1 were also found as EPO/EPOR target genes and negative-feedback components of EPO-induced erythropoiesis during anemia [25].

Many new targets have emerged in EPOR/JAK2/STAT5 signaling in erythroid cells. Singh et al. [25] found 160 EPO/EPOR target transcripts that were significantly modulated (2—21.8-fold) in primary bone marrow-derived CFUe-like progenitors. In another study, over 300 genomic locations, predominantly in promoter distal enhancer regions, were occupied by STAT5 within 30 min of EPO stimulation in murine J2E cells. Several newly discovered EPO-responsive target genes were involved in driving erythroid cell differentiation, including those involved in mRNA splicing (Rbm25), epigenetic regulation (Suv420h2), and EPOR turnover (Clint1/EpsinR) [20]. Interestingly, STAT5-binding locations linked to erythroid-related genes were co-occupied by GATA1, KLF1, and/or TAL1 transcription factor [20]. However, housekeeping genes were bound by STAT5 without GATA1 and KLF1 [21]. While STAT5 was not enriched at the promoters of the master regulator genes in J2E cells, it was enriched at the GATA1 and TAL1 promoters of human erythroleukemic K562 cells [27]. 

The occupation of STAT-binding sites seems to be cell-type- and developmental-stage-specific. That makes the signaling pathways, epigenetic, and transcriptional changes downstream of EPOR unique in different kinds of cells. For example, in the well-designed study, novel factors such as REST, an epigenetic modifier central to neural differentiation and plasticity, and NFR1, a key regulator of antioxidant response and mitochondrial biogenesis, were found in fetal neural progenitor cells. EPO treatment of these cells also altered the epigenetic landscape, identifying 1150 differentially bound histone H3 lysine 4 dimethylated regions, which are known as active and poised enhancers and transcription start sites [28]. The most recent studies revealed that the EPOR signaling axis is comprised of an exceedingly complex system of molecular signaling, epigenetic modifications, and transcriptional dynamics. The data of Schulz et al. [29] demonstrated that chromatin states vary by individual erythroid stages, with significant transition between some stages. This suggests that stage-specific transcription of genes is gradual and hierarchical process with many cis-regulatory elements included [30]. Recently, Perreault et al. [31] discovered that rapid EPO-induced transcription in murine model of erythropoiesis occurs within invariant pre-established chromatin structure. HiChIP analysis revealed chromatin contacts between H3K27ac (epigenetic modification to the DNA packaging protein Histone H3) and ubiquitous transcription factor Yin Yang 1 which was distributed to surrounding areas of EPO-regulated genes. 

In fact, evidence is emerging that specific chromatin remodeling is required for STAT binding to a subset of loci, and epigenetic regulation is a crucial and dynamic part of their gene regulation activity. Multiple studies have shown that STAT5 can also function as a transcriptional repressor by recruiting demethylating or deacetylating epigenetic modifiers to specific gene loci [32]. An experiment with EPO starvation to monitor the gene expression profile of pro-erythroblasts derived from primary fetal liver cells was carried out. In this regard, EPO induced the expression of hundreds of genes involved in the cell signaling related to the survival and identity of cells [21]. Cell identity is established and propagated primarily through epigenetic mechanisms, including the cell type-specific repertoire of enhancers [33]. The function of EPO signaling and its impact on epigenome became a high point of interest. Recently, the locations of the landscape of erythroid-specific enhancers have been determined in murine and human cell systems [34]. However, the detailed epigenetic and transcriptional patterns by which EPO signaling controls erythroid gene expression remain incompletely understood. Although transcription factors frequently bind to promoters, their predominant genome-wide distributions reside within distal enhancer regions. Enhancers are distinguished from promoters by histone modifications. Promoters are marked by histone 3 lysine 4 trimethylation (H3K4me3), while enhancer elements are demarcated by histone 3 lysine 4 monomethylation (H3K4me1) and histone 3 lysine 27 acetylation (H3K27ac) [35].

To uncover how EPO modulates the erythroid epigenome, an epigenetic profiling study was performed using an ex vivo murine cell system that underwent synchronous erythroid maturation in response to EPO stimulation [36]. Authors revealed a cis-regulatory network of thousands of EPO-responsive enhancers, of which EPO stimulation altered the histone mark signatures, highlighting EPO’s important role in reprogramming the epigenome. However, to what extent are these results comparable to humans remains unclear. Moreover, it was found that many enhancers modulated by EPO stimulation were linked to genes involved in cell-signaling pathways, such as JAK/STAT (*n* = 24), PI3K (*n* = 45), and FOXO (*n* = 25). The epigenetic landscape near genes involved in regulating the actin cytoskeleton (*n* = 31), glycosylation (*n* = 153), and transport (*n* = 221) was also altered within the first hour of exposure, suggesting that cytoskeletal reorganization may be regulated in part at the epigenetic level. 

## 3. EPO/EPOR/STAT5 Signaling in Cancer Cells

### 3.1. In Vitro Studies

Mutational analysis of the eight EPOR tyrosine residues indicated that three of them, Y343, Y460, and Y464, are required for the JAK2 V617F mutant to exhibit its oncogenic activity. Moreover, phosphorylation at these three residues was necessary for full activation of STAT5, which is a critical downstream mediator of JAK2 V617F-induced oncogenic signaling [37]. Jeong et al. [38] showed that both the prostate cancer cell lines, as well as the primary prostate tissue, express functional EPOR, which was demonstrated via the dose-dependent proliferative response to EPO and EPO-induced STAT5B phosphorylation in cells. The activation of STAT5 was also confirmed during EPO-directed suppression of apoptosis in erythroleukemic cell lines, so STAT5 was functionally implicated in the EPO-dependent survival of cells [39]. The antiapoptotic effects of EPO in both neuronal differentiated neuroblastoma SH-SY5Y and pheochromocytoma PC-12 cells require “classical” homodimeric EPOR and the combinatorial activation of multiple signaling pathways, including STAT5, AKT, and potentially MAPK, which is comparable to that observed in hematopoietic cells (Table 2) [40]. 

Although EPO stimulation of cultured EPOR-expressing breast cancer cells did not result in increased proliferation, over activation of EPOR (receptor phosphorylation) or a consistent activation of canonical EPOR signaling pathway mediators such as JAK2, STAT3, STAT5, or AKT were observed [41]. EPO-induced activation of the JAK2/STAT5, PI3K/AKT, and RAS/ERK pathways has been shown to promote malignant cell behavior in benign, non-invasive rat mammary cell lines stably transfected with human EPOR [42]. However, EPO stimulated the activation of the extracellular signal-regulated kinase p38 and c-Jun-NH(2)-kinase in MCF-7 cells but did not activate AKT or STAT5 [43]. Similarly, EPO had no effect on the JAK2/STAT5 signal transduction in MDA-MB-231 or in MDA-MB-435 cells [44].

Non-small cell lung carcinoma (NSCLC) cell line H838 expresses functional EPORs and their treatment with EPO-reduced cisplatin-induced apoptosis [45]. Another three NSCLC cell lines, A427, A549, and NCI-H358, were analyzed for the expression of EPOR and its downstream signaling pathways JAK2, STAT5, PI3K, and ERK. All cell lines expressed both mRNA and proteins of EPOR, which were found constitutively phosphorylated with downstream STAT5, AKT, and ERK1/2 independently activated of EPO treatment [46]. Additionally, pharmacological concentrations of EPO activated JAK2/STAT5, RAS/ERK, and PI3K/AKT pathways in NSCLC cells without growth advantage for these cells [47]. On the contrary, EPO mediated the invasion of head and neck squamous cell carcinoma cells through the JAK2/STAT signaling pathway (Table 2) [48].

Similarly, the co-existence of JAK2 and phosphorylated-JAK2, and STAT5 and phosphorylated STAT5, which are all involved in the mitogenic signaling of EPO, has been found frequently in the cells of malignant tumors of female reproductive organs, stomach, choriocarcinoma, melanoma, and capillary endothelial cells. Most of the phosphorylated JAK2 or phosphorylated STAT5-positive cells, however, had disappeared after the injection of monoclonal antibodies against EPO or a soluble form of EPOR [49]. In HUVEC cells, the conditioned media of hypoxic and EPO-treated A2780 cells induced significant phosphorylation of STAT5 [50]. 

**Table 2 ijms-22-07109-t002:** The list of cell lines and tumors with impact of EPOR/STAT5 signaling on the cell processes.

Cell Line/Tumor Type	Effect of EPOR/STAT5	References
Myeloproliferative neoplasms, secondary acute myeloid leukemia	Cell survival, proliferation and migration	[51]
Prostate cancer cell lines, primary prostate tissue	Proliferation	[38]
Erythroleukemic cell lines	Cell survival	[39]
Neuronal differentiated neuroblastoma SH-SY5Y, Pheochromocytoma PC-12	Cell survival	[40]
Breast cancer cells	Cell survival	[41]
Non-small-cell lung carcinoma (NSCLC) cell line H838	Cell survival	[45]
NSCLC cell lines A427, A549, and NCI-H358 cell lines	No effect	[46]
Head and neck squamous cell carcinoma cells	Tumor invasion	[48]
Multiple myeloma endothelial cells (MMEC)	Angiogenesis	[52]
Benign non-invasive rat mammary cell lines stably transfected with human EPOR	Malignant cell behavior	[42]

### 3.2. In Vivo Studies and Patients

To date, epoetin alpha, beta, and darbepoetin alpha have been approved for anemia treatment in cancer patients. Although epoetins act on the same EPOR, differences in pharmacodynamics and pharmacokinetics through variations in glycolylation degree are expected. However, though both epoetins alpha and beta demonstrated comparable efficacy and safety profiles, epoetin beta achieved targeted hematocrit levels with lower doses [53] and kept the occurrence of hemoglobin levels above 13 g/dL in EPO-treated end-stage renal disease hemodialysis patients [54]. 

Under physiological conditions, STAT5 expression is regulated by prolactin, growth hormone, insulin growth factor, estrogen, and progesterone signaling pathways, and plays a significant role in a normal mammary cell differentiation, proliferation, and survival [55]. Watowich et al. [56] have shown a lack of specificity of STAT signaling during red blood cell development. However, the importance of STAT5A/B proteins in the erythroid differentiation activated by EPO/EPOR/JAK2 signaling has been well demonstrated [20,21]. Although deletion of erythroid STAT5A/B resulted in anemia [57], Villarino et al. [58] found no difference in the hematocrit of single-allele expressing mice. STAT5A/B are upregulated in myeloproliferative neoplasms, acute lymphoblastic leukemia (ALL), chronic myelogenous leukemia (CML), and B-cell and peripheral T-cell leukemia/lymphoma [51]. Increased STAT5A/B activation is achieved by gains in copy number, by upregulation of protein expression, or gain-of-function (GOF) mutations, which can lead to a higher phosphorylated form of STAT5A/B levels (pYSTAT5A/B) followed by tumor cell survival and disease progression [5]. The research in mature NK/T-cell neoplasms is currently focused on the recurrent point mutations localized mainly in the SH2 domain of STAT5B. The most frequent mutation, STAT5B^N642H^, was observed in many forms of peripheral T-cell leukemia/lymphomas [59,60,61] and has also been detected in myeloid neoplasia with eosinophilia [62] and neutrophilic leukemia [63]. Moreover, a transgenic mouse model with STAT5B^N642H^, under the hematopoietic vav-promoter, developed an aggressive CD8+ T-cell leukemia with organ infiltrations by CD8+, CD4+, and γδ T-cells [64,65]. The oncogenic role for STAT5B^N642H^ was confirmed by a transplantation model derived from the transgenic mouse in NKT [66] and γδ T-cells [65] mentioned above. 

Bone marrow-derived endothelial cells from monoclonal gammopathy (MGEC) of undetermined significance and multiple myeloma endothelial cells (MMEC) were assessed for the possibility that EPO might play a role in MGECs- and MMECs-mediated angiogenesis. Lamanuzzi et al. [52] showed that EPOR is expressed in both MGECs and MMECs, and at higher levels in the former (Table 2). Both EC types respond to recombinant human EPO (rHuEPO) in terms of cell proliferation and activation of JAK2/STAT5 and PI3K/AKT pathways. On the other hand, bone marrow-derived dendritic cells, which are direct targets of EPO, have demonstrated STAT3 rather than STAT5 activation [67]. Interestingly, the overexpression of HIF-1a, with stimulated EPO expression, diminished ROS levels and weakened cell damage caused by the iron overload in myelodysplastic syndrome patients [68]. Similarly, iron chelation therapy reduced the expression of SOCS1, recovered EPOR/STAT5 signalization, and reduced the apoptosis of immature erythrocytes in the bone marrow of low-risk myelodysplastic syndrome patients [69]. 

Similar signal transduction, including the STAT5 pathway, is mediated via EPOR-βcR, although EPOR splice variants [70] and EPOR-βcR heterodimers [71] have received much less consideration. The only difference in EPOR-βcR signaling compared with EPOR homodimer is the high concentration of EPO required. Ephrin-type B-receptor 4 (EPHB4) was also identified as an alternative EPOR that triggers downstream signaling via STAT3 and promotes recombinant human EPO-induced tumor growth and progression [72]. 

## 4. EPO/EPOR/STAT5 Induced Protection of Normal Cells

Protection mediated by EPO/EPOR/JAK2 results in pancreatic β cell proliferation and survival, increased angiogenesis, and reduced inflammation in the islets [73], as well as impaired pathogen clearance due to reduced production of anti-microbial TNF-α and NO molecules [74]. Recently, Li et al. [75] demonstrated that kidney-secreted cytokine EPO controls blood lipids in kidney disease. EPO has been shown to prevent necrotic ischemic injury and cisplatin-induced nephrotoxicity in primary mouse renal tubular epithelial cells [76] and human renal proximal tubular epithelial cells [77], possibly via JAK2/Y343/STAT5 signaling and Pim3 as a target gene of STAT5 [76]. EPO induced JAK2/STAT5 phosphorylation and nuclear translocation of STAT5, without the activation of MAPK38α and ERK1/2, followed by pronounced release of miRNA-210 from peripheral blood mononuclear cells in vitro. Hemodialyzed end-stage renal disease patients also demonstrated EPO-stimulated miRNA-210 as a result of JAK2/STAT5 signal transduction in peripheral blood mononuclear cells. Whereas some miRNAs can be used as a marker of urological [78] or prostate [79] cancer diagnosis, plasmatic miRNA-210 negatively correlates with the EPO resistance index and is considered to be a prognostic marker of EPO responsiveness and an index for anemia management in hemodialyzed end-stage renal disease patients [80].

STAT5 is also required for lipid degradation and its deficiency supports adiposity and impairs lipid mobilization in the adipocytes of mice [75,81,82]. In this regard, EPO stimulated lipid catabolism through the activation of JAK2/STAT5 signaling in peripheral adipose tissue and ameliorated non-alcoholic fatty liver disease via EPO/EPOR-induced STAT3 and STAT5 activation [83]. EPO and darbepoetin alpha improved glucose tolerance and insulin resistance, inhibited lipid accumulation in the liver and white adipose tissue, and reduced body weight. Their administration led to downregulation of proteins involved in the synthesis of lipids in the liver, such as fatty acid synthase, acetyl-CoA carboxylase, and sterol regulatory element-binding protein 1. They additionally led to upregulation of proteins involved in the lipolysis of visceral white adipose tissue, such as anti-adipose triglyceride lipase and hormone sensitive lipase [83,84]. In the brown fat tissue of young male mice, EPO therapy regulated differentiation of brown adipocyte, STAT3 activation, secretion of FGF21, and improved insulin sensitivity and glucose tolerance via upregulation of master transcriptional coregulator PRDM16 [84]. Recently, Suresh et al. [85] demonstrated that endogenous EPO signalization modulates differentiation of bone marrow stromal cells, and yet the aberrant EPO reduces osteogenesis and favors adipogenesis. It therefore seems that EPO plays a significant role in osteoblastosis and is important for both normal bone formation and bone loss during EPO-induced erythropoiesis [86,87,88]. Bone loss could be partly mediated by EPO targeted B-cells via both increased expression of osteoclastogenic molecules, as well as by transdifferentiation of B-cells into osteoblasts [89]. 

The upregulation of STAT5 and its downstream gene products, Bcl-xL and XIAP, demonstrated significant protective effects on astrocytes (during 24–48 h anoxia) and A9 dopaminergic neurons of the substantia nigra [90]. Both effects were attenuated by inhibition of JAK2/STAT5 signaling [91]. In cultured cerebrocortical neurons, EPO/EPOR-triggered phosphorylation of STAT5 has been shown to activate nuclear factor κB (NF-κB) and subsequently induce expression of the antiapoptotic proteins X-linked inhibitor of apoptosis (XIAP) and c-inhibitor of apoptosis-2 (cIAP2), which in turn suppress caspase 3, 8, and 9 [92]. Interestingly, EPO-mediated neuroprotection involves cross-talk between JAK2 and NF-κB signal cascades [93]. Although not required for neuroprotection of EPO in hippocampal neurons, STAT5 is essential for EPO-neurotrophic activity [94]. In the detached retina, EPO activates both PI3K/AKT and MAPK/ERK-1/2 signal transduction pathways. Intravitreal injection of EPO has been demonstrated to protect both retinal vascular and neuronal cells in early diabetes via the activation of ERK but not the STAT5 pathway [95]. Conversely, EPO reduces insulin resistance and increases glucose uptake in adipocytes through the activation of AKT and STAT5 [96]. 

In rats with experimental epilepsy, EPO protected myocardial cells from apoptosis via the JAK2/STAT5 pathway, and the inhibition of JAK2/STAT5 signalization by AG490 reduced the antiapoptotic effects of EPO [97]. On the other hand, carbamyl-EPO variants and pharmacological doses of EPO (250 IU/mL) exerted antiapoptotic activity in myocardial cells [98] and vascular smooth muscle cells (VSMCs) [99], respectively, via a pathway independent of JAK2/STAT5 signaling. In addition, modulating activities of EPO on adaptive and innate cells of immune systems are reviewed in Cantarelli et al. [100].

## 5. Conclusions

Recently, several new target genes of the EPO/EPOR/STAT5 signaling cascade have been discovered, especially in erythroid cells. These genes and their protein products are mainly involved in the processes of epigenetic regulation, mRNA splicing, molecular signaling, EPOR turnover, and negative regulation of STAT5 signaling. However, they can also be implicated in the regulation of cell metabolism and cytoskeletal reorganization. Transcriptomic analyses and mapping of EPO/EPOR target genes are also necessary for tumor and normal cells, on which only signaling and the effect of EPO/EPOR have been described so far, e.g., protective, antiapoptotic, trophic, and other. New EPO/EPOR target genes in cancer cells could serve as markers for various means of monitoring both disease activity and cell responses to various therapies, including the administration of rHuEPO to tumor patients.

## 6. Source of Data 

We reviewed most recent scientific papers and the data from the biomedical literature found in PubMed. Studies were obtained using the following keywords: “STAT5” or “erythropoietin” or “erythropoietin receptor” and “erythropoiesis” or “cancer cell” or “normal cell” or “signaling”. 

## Figures and Tables

**Figure 1 ijms-22-07109-f001:**
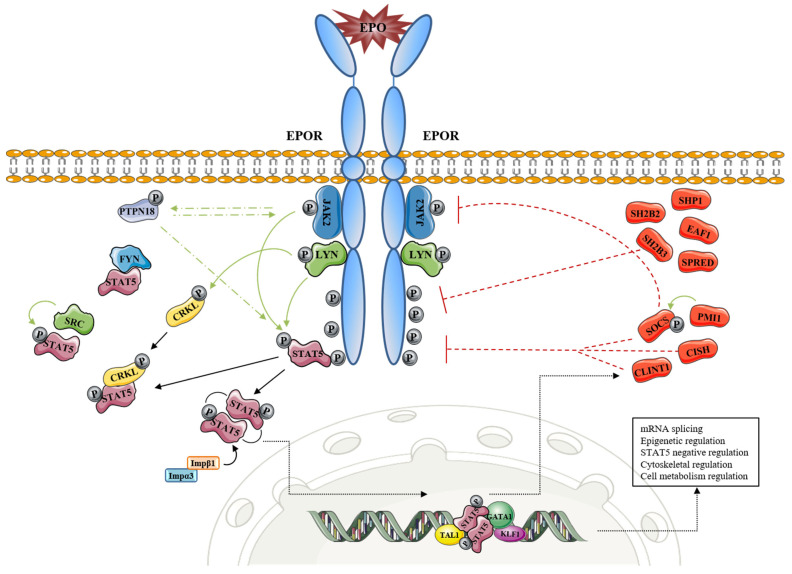
EPOR-STAT5 signaling and its involvement in processes other than inhibition of apoptosis, cell cycle progression, and cell cycle arrest. Green line: activation effect (phosphorylation); red line: inhibition effect; black line: transition.

**Table 1 ijms-22-07109-t001:** Main players of JAK2/STAT5 signaling cascade.

Gene	Gene ID	Protein Name	Mechanism	Reference
*JAK2*	3717	Tyrosine-protein kinase JAK2	EPORactivation	[16]
*STAT5A*	20850	Signal transducer and activator of transcription 5A	Signal transduction and activation of transcription of genes of erythroid differentiation	[5]
*STAT5B*	6777	Signal transducer and activator of transcription 5B		
*LYN*	4067	Tyrosine-protein kinase Lyn	Phosphorylation/activation of STAT5 Phosphorylation/activation of CRKL	[13]
*SRC*	6714	Proto-oncogene tyrosine-protein kinase Src	Phosphorylation/activation of STAT5	[11]
*FYN*	2534	Tyrosine-protein kinase Fyn	Phosphorylation/activation of STAT5	[17]
*CRKL*	1399	CRK-Like protein	Phosphorylation/activation of STAT5	[14,18]
*PTPN18*	26469	Tyrosine-protein phosphatase non-receptor type 18	Activation of STAT5	[15]
*KPNA4*	3840	Importin subunit alpha-3	Nuclear import/export	[9,19]
*KPNB1*	3837	Importin subunit beta-1		
*XPO1*	7514	Exportin-1 (CRM1)		
*GATA1*	2623	Erythroid transcription factor	Regulation of STAT5 target gene transcription Transcription factor	[20]
*KLF1*	10661	Krueppel-like factor 1	Regulation of STAT5 target gene transcription Transcription factor	[20]
*TAL1 (SCL)*	6886	T-cell acute lymphocytic leukemia protein1	Regulation of STAT5 target gene transcription Transcription factor STAT5 target gene	[21]
*BCL2L1*	598	Bcl-2-like protein 1 (apoptosis regulator Bcl-X)	STAT5 target gene	[20]
*CLINT1*	9685	Clathrin interactor 1 (EPSINR)	STAT5 target gene EPOR internalization & negative feedback for EPO-induced signaling	[20]
*CISH*	1154	Cytokine-inducible SH2-containing protein	STAT5 target gene Inhibition of STAT5 signaling Degradation of EPOR	[20,22,23]
*SOCS1*	8651	Suppressor of cytokine signaling 1	STAT5 target gene JAK2 degradation Inhibitor of EPO signaling/Inhibition of STAT5 signaling	[22]
*SOCS2*	8835	Suppressor of cytokine signaling 2	STAT5 target gene Inhibition of STAT5 signaling	[22]

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
