# Peer review of "STAT5 as a Key Protein of Erythropoietin Signalization"

_ijms, 2021, doi:10.3390/ijms22137109_

Round 1

Reviewer 1 Report

Review on the manuscript titled “STAT5 as a key protein of erythropoietin signalisation” by Zuzana Tóthová, Jana Tomc, Nataša Debeljak, Peter Solár, submitted to Int. J. Mol. Sci.

In the manuscript, the authors quite accurately describe the intracellular effects of erythropoietin. The authors broadly reviewed transcriptomics, proteomics, and epigenetic studies facilitating the understanding of EPO signaling.

Please reconsider the following:

  1. The research methodology for article identification and selection, keywords reporting, screening, and rated articles should be presented.
  2. It would be desirable if the chapter on EPO/EPOR/STAT5 signaling in cancer cells were divided into a section on in vitro and in vivo studies.
  3. Since different eytropoietins (α and β) are used in therapy it might be useful to outline the differences between them.

Author Response

 the following: 

  1. The research methodology for article identification and selection, keywords reporting, screening, and rated articles should be presented 

.Thank you. We have added additional chapter “Source of data”, explaining the methodology and     the system of papers selection. 

  1. It would be desirable if the chapter on EPO/EPOR/STAT5 signaling in cancer cells were divided into a section on in vitro and in vivo studies. 

Thank you for suggestion. We have accepted this point and divided EPO/EPOR/STAT5 chapter       into two sections. 

  1. Since different eytropoietins (α and β) are used in therapy it might be useful to outline the differences between them. 

Thank you. New information about different epoetins at the beginning of chapter 3.2. were added. 

Reviewer 2 Report

The review written by Tothova et al, entitled « Stat5 as a key protein of erythropoietin signalisation » is very interesting and complete.

Main point

  • There are only one Figure and one Table in the review and it is for the first point of the review « 1-STAT5 signaling and erythropoiesis ».
  • The authors should add some figues, by instance in the point « 2. The interplay of EPO signalling, transcription patterns an epigenetics », a figure with the different signalling pathways could help for understanding. A table for point « 3.EPO/EPOR/STAT5 signaling in cancer cells could be added too.
  • Lines 315-321, the authors should clarify their purpose, because it is difficult to understand the link between this sentences.

Minor points

  1. In line 140, the authors should define H3K27ac and Yin Yang.

Author Response

he review written by Tothova et al, entitled « Stat5 as a key protein of erythropoietin signalisation » is very interesting and complete. 

Main point 

  • There are only one Figure and one Table in the review and it is for the first point of the review « 1-STAT5 signaling and erythropoiesis ». 
  • The authors should add some figues, by instance in the point « 2. The interplay of EPO signalling, transcription patterns an epigenetics », a figure with the different signalling pathways could help for understanding. A table for point « 3.EPO/EPOR/STAT5 signaling in cancer cells could be added too. 

Thank you for suggestion. We have added new table: The list of cell lines and tumors with impact of EPOR/STAT5 signaling on the cell processes. Regarding EPOR signaling, we think that our  figure 1 with EPOR/STAT5 signaling in the combination with table 1 shows the most important positive and negative EPOR/STAT5 signals and the addition of other images could be   sweeping.  

  • Lines 315-321, the authors should clarify their purpose, because it is difficult to understand the link between this sentences. 

Thank you for suggestion. We have reformulated mentioned sentences and we hope whole paragraph is more understandable now.  

 Minor points 

  1. In line 140, the authors should define H3K27ac and Yin Yang. 

Thank you. Both term are explained in the text now. 

Round 2

Reviewer 2 Report

The authors made the modifications I asked.